# Tracking the Transcription Kinetic of SARS-CoV-2 in Human Cells by Reverse Transcription-Droplet Digital PCR

**DOI:** 10.3390/pathogens10101274

**Published:** 2021-10-02

**Authors:** Ka-Ki Au, Chunke Chen, Yee-Man Chan, Winsome Wing Sum Wong, Huibin Lv, Chris Ka Pun Mok, Chun-Kin Chow

**Affiliations:** 1Medtimes Molecular Laboratory Limited, Hong Kong, China; kitty@medtimes.com.hk (K.-K.A.); ymchan@gmail.com (Y.-M.C.); wongwsw@hotmail.com (W.W.S.W.); 2The Jockey Club School of Public Health and Primary Care, Faculty of Medicine, The Chinese University of Hong Kong, Hong Kong, China; chunkechen@cuhk.edu.hk; 3Li Ka Shing Institute of Health Sciences, Faculty of Medicine, The Chinese University of Hong Kong, Hong Kong, China; 4HKU-Pasteur Research Pole, Li Ka Shing Faculty of Medicine, The University of Hong Kong, Hong Kong, China; huibin01@connect.hku.hk

**Keywords:** coronavirus, SARS-CoV-2, digital droplet PCR, transcription, subgenomic RNA

## Abstract

Viral transcription is an essential step of SARS-CoV-2 infection after invasion into the target cells. Antiviral drugs such as remdesivir, which is used to treat COVID-19 patients, targets the viral RNA synthesis. Understanding the mechanism of viral transcription may help to develop new therapeutic treatment by perturbing virus replication. In this study, we established 28 ddPCR assays and designed specific primers/probe sets to detect the RNA levels of 15 NSP, 9 ORF, and 4 structural genes of SARS-CoV-2. The transcriptional kinetics of these viral genes were determined longitudinally from the beginning of infection to 12 h postinfection in Caco-2 cells. We found that SARS-CoV-2 takes around 6 h to hijack the cells before the initiation of viral transcription process in human cells. Our results may contribute to a deeper understanding of the mechanisms of SARS-CoV-2 infection.

## 1. Introduction

There have been more than 179 million COVID-19 cases around the globe and nearly 4 million associated deaths since the initial disease outbreak in late 2019 [1]. The causative agent that is responsible for the disease is a new and emerging strain of coronavirus, namely SARS-CoV-2 [2,3]. SARS-CoV-2 belongs to the beta group of coronaviruses, which is in the family of Coronaviridae. The architecture of SARS-CoV-2 has been studied in detail [4]. Inside each virion, there is a positive sense, single-stranded viral genome which is around 30 kb long. Its genome encodes for four major structural proteins: spike (S), envelope (E), matrix (M), and nucleocapsid (N); sixteen nonstructural proteins (nsp1 to nsp16); and ten accessory proteins (ORF1a/1ab, 3a/b, 6, 7a/b, 8, 9b, and 10) [4].

SARS-CoV-2 mainly replicates in the human respiratory tract after infection [5]. After invading into the target cells, the SARS-CoV-2 virus initiates translation of two major replicase polyproteins, the pp1a (ORF1a) and pp1ab (ORF1ab) from the ORF region of its positive sense RNA genome. Most of the transcription- and translation-dependent proteins such as papain-like proteases, 3C-like protease, helicase, and RNA-dependent RNA polymerase are encoded at these regions [6]. Formations of the replication and transcription complex further drive the transcription of subgenomic mRNAs (sgRNAs) and new viral RNA genome. The sgRNAs are responsible for the production of various viral proteins. Other than the structural proteins, it is known that the nonstructural proteins (NSP) and the open reading frame (ORF) proteins are also essential during the replication process. At the late stage of replication cycle, the translated structural proteins are translocated into endoplasmic reticulum (ER)/Golgi and assembled into a virion with the newly produced genomic viral RNA. The virion is then finally released from the cell by exocytosis [7]. 

The entire replication cycle of a coronavirus takes more than ten hours [8,9]. Although some studies has sought to determine the transcriptional profile of SARS-CoV-2 at selected time points of postinfection, the kinetic of the viral transcription in human cells is still yet to be completely resolved [10,11,12]. Droplet digital PCR (ddPCR) is recently used to quantify low abundance of SARS-CoV-2 subgenomic viral gene transcripts by its advantage of low template requirementfor the reaction [13]. Ultra sensitive clinical diagnosis is one of the major applications of ddPCR, in such case the assay can identify the COVID-19 patients through detecting low expression of viral genes from their specimens [14]. However, the concern is that the primers and probes used for the current detection may not be updated promptly, as mutations are frequently identified from the new variants of SARS-CoV-2 [15]. In this study, we designed primers/probes sets for the ddPCR that target different viral genes of SARS-CoV-2. The ddPCR assays were then used to track the transcriptional kinetic of the 15 NSP, 9 ORF, and 4 structural genes of SARS-CoV-2 during the initial replication cycle in Caco-2 cells, which is a human cell line that is susceptible for the replication of SARS-CoV-2.

## 2. Materials and Methods

### 2.1. SARS-CoV-2 Sequences and Alignment

On 9 September 2020, 61,013 SARS-CoV-2 genome sequences were downloaded from GISAID, followed by MAFFT sequence alignment. Mutations from each viral gene were identified through the comparison to the sequence of BetaCoV/Hong Kong/VM20001061/2020 and 50 nucleotide positions with the highest mutation frequencies from each gene were determined. The viral gene mutation coordinates and their frequencies were annotated and summarized in Appendix A.

### 2.2. Primers and Dual-Labeled Hydrolysis Probes

All the primer oligos and dual-labeled fluorescent probes were synthesized and purified by Life Technologies and Sangon Biotech (Shanghai). Human ribonuclease subunit p30 (RNase P) was used to serve as our endogenous control.

### 2.3. Viruses and Cells

The SARS-CoV-2 virus strain (BetaCoV/Hong Kong/VM20001061/2020) was propagated in Vero E6 cells and the infectious titer of the viral stock was determined by serially diluting the virus on the Vero E6 cells by plaque-forming unit (pfu). All the experiments of virus culture were carried out in the biosafety level 3 containment facility in the University of Hong Kong and fully in accordance with the laboratory biosecurity and biosafety guidelines. 

Human Caco-2 cell lines were purchased from ATCC and grown in a T-75 flask (Greiner Bio-One CELLSTAR, Austria) with Dulbecco’s modified eagle medium (DMEM) that was supplemented with 10% fetal bovine serum (FBS) (GIBCO, USA), 2 mM HEPES (Gibco), 100 U/ml of penicillin, 100 µg/ml of streptomycin, and 1% of GlutaMax (Gibco, USA) and until 90% confluency. Cells were then dissociated with trysin-EDTA (GIBCO, USA) and seeded into a 24-well tissue culture plate (TPP, Switzerland) at a concentration of 0.5×10^6^ cells per well. Culture medium was then changed to 0% FBS–DMEM during and after the infection. All the cell cultures were incubated and grown at 37 °C and maintained with 5% CO_2_ in the incubator.

### 2.4. Virus Infection and Collection of Cell Lysate

Caco-2 cells were mock-infected or infected by SARS-CoV-2 at multiplicity of infection (MOI) of 0.01. After 15 min, the cells were either lysed by 350 µL of RNA lysis buffer (Buffer RLT, Qiagen, Germany) after washed twice with pre-warmed 1× PBS (T = 0) or further incubated for additional 45 min. At 1-h postinfection (hpi), the cells were washed with pre-warmed 1× PBS and replaced with fresh cell-culture medium (DMEM, 0% FBS). The infected cells were further incubated in 37 °C incubator and the total RNAs were then harvested by the RNA lysis buffer at 2, 4, 6, 8, 10 and 12 hpi. All cell lysates were then kept at −80 °C until RNA extraction.

### 2.5. Preparation of cDNA Templates 

Total RNAs were extracted from the cell lysates by using the RNeasy Mini Kit (QIAGEN, Germany) following the manufactory’s protocol. In brief, 350 µL of the cell lysates containing buffer RLT were input for extraction and purified RNAs were eluted in 50 µL of RNase-free water. All RNAs extracted were then stored at −80 °C until use. Reverse transcription was performed to generate cDNA from total RNA by using the LunaScript® RT SuperMix Kit (BioLabs, New England). In brief, 16 µL of the purified RNAs were mixed with the 4 µL of the 5× reaction mix and incubated at 25 °C for 2 min for primer annealing, followed by cDNA synthesis at 55 °C for 10 min and enzyme inactivation at 95 °C for 1 min. cDNAs were then subsequently diluted in 1:20 with 1× TE buffer, pH 8.0 (Sigma-Aldrich, USA) and were kept at −20 °C until use.

### 2.6. Endogenous Control and Absolute Quantification by ddPCR

The copy numbers of the viral genes and the endogenous RNase P were absolutely quantified in the ddPCR system. To co-amplify the viral genes and endogenous control (RNase P gene) in the same reaction, we applied duplex TaqMan probes for the ddPCR and labeled them with 6-FAM and VIC fluorescent signals respectively [16]. In brief, 20 µL of a PCR reaction mix that contains 10 µL of 2× ddPCR Supermix for Probes (No dUTP), 2 µL of a primer/probe mixture (final concentration of each primer: approximately 900 nM; hydrolysis probe: approximately 250 nM), 3 µL of 1:20 diluted cDNA, and 5 µL of RNase–DNase-free water was prepared. The PCR reaction mix was then partitioned with droplet generation oil in a QX200™ Droplet Generator for droplets generation. The partitioned products (roughly 40 µL) were then transferred to a new 96-well PCR plate (0.2 mL) and amplified in a C1000 Touch Thermo Cycler by using the following cycling conditions: enzyme activation at 95 °C for 10 min, followed by 40 cycles of a two-stage-amplification at 94 °C for 30 s, at 60 °C for 1 min, and finally at 98 °C for 10 min. The partitioned droplets containing end-point fluorescent-labeled PCR products were then quantified immediately by QX200 Droplet Reader. Gating of the FAM/VIC counts was performed in QuantaSoft software according to the manufacturer instructions.

### 2.7. Validation of the Primers and Probes on Clinical Specimens by RT-QPCR

We selected 12 combined nasopharyngeal and throat swab clinical specimens that were previously laboratory confirmed as SARS-CoV-2-positive for the study. In brief, the viral RNA was extracted from 140 µL of viral transport medium by using the QIAamp Viral RNA Kit (QIAGEN, Germany) that following the manufacturer’s protocol and purified in 50 µL of buffer AVE. The synthesis of cDNA was performed as described in Section 2.5. The cDNAs were then diluted in 1:10 with 1× TE buffer, pH 8.0 (Sigma-Aldrich, USA), and kept at −20 °C until use. Quantitative real-time PCR was then performed by using the following sets of primers and probes: S, E, M, N, orf1a, orf1b, nsp1, nsp2, nsp3, nsp4, nsp5, nsp6, nsp7, nsp8, nsp9, nsp10, nsp12, nsp13, nsp14, nsp15, nsp16, orf3a, orf6, orf7a, orf7b, orf8, orf9b, orf10. Informed and written consents were obtained from all participants and the study was approved by the Medtimes Medical Group Ethics Review Board.

In brief, 10 µL of reaction mix containing 5 µL of 2× PerfeCTa qPCR ToughMix (Quantabio, USA); 1.5 µL of primer and probe mixture that comprised 400 nM each of the forward and reverse primers and 200 nM each of the fluorescent hydrolysis probes; 3 µL of 1:10 diluted cDNA; and 0.5 µL of rox reference dye was prepared for each reaction. The qPCR reaction mix was then transferred to a 0.1 mL 96-well qPCR plate (Applied Biosystems, USA). The viral gene templates were amplified and the fluorescent signal was acquired by the ViiA 7 Real-Time PCR System (Thermo Fisher Scientific, USA) using the following cycling conditions: initial denaturation at 95 °C for 1 min, followed by 45 cycles of a two-stage amplification at 95 °C for 2 s and at 60 °C for 12 s. QuantStudio™ Real-Time PCR Software v1.6.1 was then used for data analysis.

### 2.8. Data and Statistical Analysis

The one-way ANOVA statistics model was applied by using GraphPad Prism 9 (GraphPad Software Inc) and the viral gene copies were normalized with endogenous mRNA control of ribonuclease P protein subunit p30 gene (RNase P gene). The absolute counting strategy is being enumerated and listed below, while FAM signal represented the target viral genes, VIC signal represented the endogenous control, and FAM/VIC heterogeneous signal reflected the droplets containing both signals and that were being co-amplified.
(1)$$\mathrm{Viral}\;\mathrm{RNA}\;\mathrm{gene}\;\mathrm{counts}\;\mathrm{per}\;\mathrm{copy}\;\mathrm{of}\;\mathrm{RNase}\;P\;\mathrm{gene}=\frac{\left(\mathrm{FAM}\right)+\left(\mathrm{FAM}/\mathrm{VIC}\right)}{\left(\mathrm{VIC}\right)+\left(\mathrm{FAM}/\mathrm{VIC}\right)}$$

To address the concern of amplification bias by using different primer/dual-labeled hydrolysis probe sets and resulting the counting discrimination, we additionally designed three sets of primers and probes on another conserve regions of nsp3 (B), S gene (B), and N gene (B). This aimed to determine the consistency of different primer/probe sets that targeting the same genes. Two-way ANOVA statistical analysis model was applied to assess the variations of primers and probes used. The difference was determined as significance when *p* < 0.05.

## 3. Results

There are totally 16 nonstructural proteins (nsp 1 to 16), 4 structural proteins (S, E, M, and N) and 10 accessory proteins (orf1a/ab, 3a/b, 6, 7a/b, 8, 9b, and 10) encoded by the viral genome of SARS-CoV-2 (Figure 1). To identify suitable target regions for the ddPCR, 61,013 of SARS-CoV-2 sequences were downloaded from GISAID and aligned. Twenty-eight sets of primers and dual-labeled hydrolysis probes that targeting the conserved regions of different viral genes, including 4 structural proteins, 15 nonstructural proteins, and 9 accessory proteins, were designed (Table 1). The ddPCR for Nsp11 and orf3b genes were not included due to their short coding lengths. Oligos for the orf1a and orf1ab were designed to target the overlapping regions with nsp 2/3 and nsp 12/13 respectively. Additionally, two sets of primers and probes were designed to target to the 5′ and 3′ untranslated regions (UTR) respectively. To analyze the transcription pattern of subgenomic viral RNAs (sgRNA) and understand how the leader sequences are fused to the open reading frames, three additional sets of primers and probes were designed to target the 1) Leader–TRS, 2) Leader–TRS–N gene and 3) orf10-3′ UTR respectively.

Human Caco-2 cells were infected by the SARS-CoV-2 and the total RNAs were collected at 15 min, and 2, 4, 6, 8, 10, and 12 h postinfection. The transcription levels of the viral genes were determined by ddPCR using the corresponding primers and probe. In general, the kinetics of the viral transcription pattern were similar among all viral genes (Figure 2A–C). There was a significant decrease of the viral RNA level from the beginning of the infection to 2 h postinfection. No significant change of viral transcription was found from 2 to 6 h after infection, while dramatic increase of the viral RNA was observed beyond 6 h postinfection. Similarly to other coronaviruses, transcription of N, Orf9b, and Orf10 were the most abundantly expressed among all viral genes encoded by the genome. This can be explained by the fact that their encoding regions are closer to the 3′ end of the viral genome than the other genes. Interestingly, we found no difference in the transcription among all NSP genes (Figure 2C). These results support that all the NSPs share one subgenome for their transcription. Reproducibility and performance of our ddPCR assays were further evaluated by using alternative sets of primers/probes that target different encoding regions of nsp3, N, and S. We showed that the alternative sets of primers/probes (nsp3 (B), N (B), and S(B)) did not lead to significant variation of our ddPCR results (Appendix A).

Discontinuous viral transcription process is a hallmark of coronavirus that produces a set of nested 3′and 5′ co-terminal subgenomic RNAs for different viral genes. Since detection of N gene using our primer/probe set may also represent the subgenomic RNAs that are used for the transcription of other viral genes, we then sought to estimate the proportion of the N gene from the total transcription. We first determined the level of the total transcription from the infection using a primers/probe set which specifically targets the Leader–TRS region or 3′UTR. The transcription that is specific for N gene was then detected by another set of primers/probe that covers the regions of Leader, TRS, and N. We found that about 38.1–39.6% and 53.4–56.7% among the total transcription involving the Leader–TRS and 3′UTR are specific for the transcription of N gene respectively (Figure 3A). Recently, a study reported that the protein expression of pp1a is 1.4–2.2 times higher than pp1ab. We found that the transcription levels of the ORF1a and ORF1b are similar, which support the hypothesis that the cause of the difference in the protein expression may be due to stoichiometry (Figure 3B).

To further determine the performance of the primers/probe, the expression levels of the structural, NSP, and ORF genes from 12 clinical swab specimens were tested by real-time qPCR (Table 2). All of the target genes were able to be amplified by our primers/probe and detected by the assay. Similarly to the results of our in vitro experiments, N and ORF9b genes showed the highest level of expression among all the target genes in each specimen.

## 4. Discussion

Compared to traditional quantitative PCR (qPCR), ddPCR is a more sensitive assay for the detection of low levels of gene expression. Our ddPCR assays have the potential to be used for contact tracing so that COVID-19 patients can be identified during the early phase of their infections. The evolution of SARS-CoV-2 since the beginning of the outbreak has resulted in the emergence of variants of concern (VOCs) [15]. Some mutations such as the deletion at position 69/70del can cause a mismatch on the primer, which is used to target the S gene [15]. Moreover, the use of newly discovered drugs such as remdesivir or simeprevir may also cause escape mutations at the orf regions. Thus, these mutations may affect the accuracy of the diagnosis from detecting the viral nucleic acid. The primers and probes that we designed for detecting SARS-CoV-2 cover 15 NSP, 9 ORF, and 4 structural protein genes. All of them were designed and kept away from the highly variable positions of the SARS-CoV-2 genome and we expect that they may be useful for new variants detection in the coming future. 

While the efficacy of remdesivir in humans is still suboptimal, structurally modification of this drug or identification of new compound will be one of the key directions for antiviral research. As we have demonstrated the kinetic of the transcription process of the SARS-CoV-2 in Caco-2 cells, our model may be useful for investigating the specific functions of new antiviral drugs such as delaying the incubation time before the initiation of viral transcription or reducing the transcription level, etc. Our study thus provides a model for evaluating the performance of any new antiviral drugs against the SARS-CoV-2 and their mechanism of action intensively.

The results from our infection model in human Caco-2 cells using digital droplet PCR assay has tracked the kinetic of the transcription profile of SARS-CoV-2 at the early cycle of replication. Although some studies also measured the transcription profile of this virus, primate origin cell lines, such as Vero cells, were used, which may not be physiologically relevant to the transcription property of SARS-CoV-2 in humans [10,11]. On the other hand, we traced the change of the viral transcription at every 2 h following 15 min after the infection. The results thus can provide a clear picture on how the viral transcription is being regulated during the first virus life cycle. 

There was an obvious decrease of viral RNA at 2 h postinfection, suggesting that the input viral RNA was consumed for the translation after the entry step of SARS-CoV-2. It is known that the positive strand nature of the SARS-CoV-2 genomic RNA enables the virus to translate its own replicase–transcriptase-complex (RTC) by using host cell ribosomes [6]. Here, our data showed that SARS-CoV-2 requires about 6 h to hijack the host transcription machinery before it can further transcribe its subgenomic RNAs. We also found that there was absence of productive viral transcription in Caco-2 cells between 2 and 6 h postinfection. The study from Hofmann et al. showed that the viral transcription of bovine coronavirus (BCoV, beta-coronavirus) started at 3–4 h postinfection in human rectal tumor (HRT) cells [17]. The reason for the different transcription kinetics between the two coronaviruses will need to be further investigated. Moreover, it will be desirable to further explore the virology of the SARS-CoV-2 using primary lung epithelial cells or ex-vivo organoids.

## 5. Conclusions

In this study, we established 28 ddPCR assays with specific primers/probe sets to detect the transcription profiles of 15 NSP, 9 ORF, and 4 structural protein genes of SARS-CoV-2. The transcriptional kinetic of the viral genes of SARS-CoV-2 during the initial replication cycle in human cell was determined. We also found that SARS-CoV-2 takes around 6 h to hijack the cells before initiating a productive viral transcription process.

## Figures and Tables

**Figure 1 pathogens-10-01274-f001:**
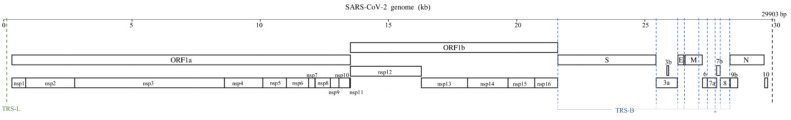
Genomic arrangement and coordinates of SARS-CoV-2. The encoding regions of different viral genes of SARS-CoV-2 (29,903bp, NC_045512) are shown. There are 16 nonstructural proteins (nsps 1 to 16) encoded by orf1a and orf1b, 4 structural proteins (S, E, M, and N), and 8 accessory proteins (orf3a/b, 6, 7a/b, 8, 9b, and 10). The transcription regulatory sequences (TRS) that is located after the Leader sequences (TRS-L) is highlighted in green dash lines. The TRS that is located before each individual open reading frame (TRS-B) are highlighted in blue dash lines.

**Figure 2 pathogens-10-01274-f002:**
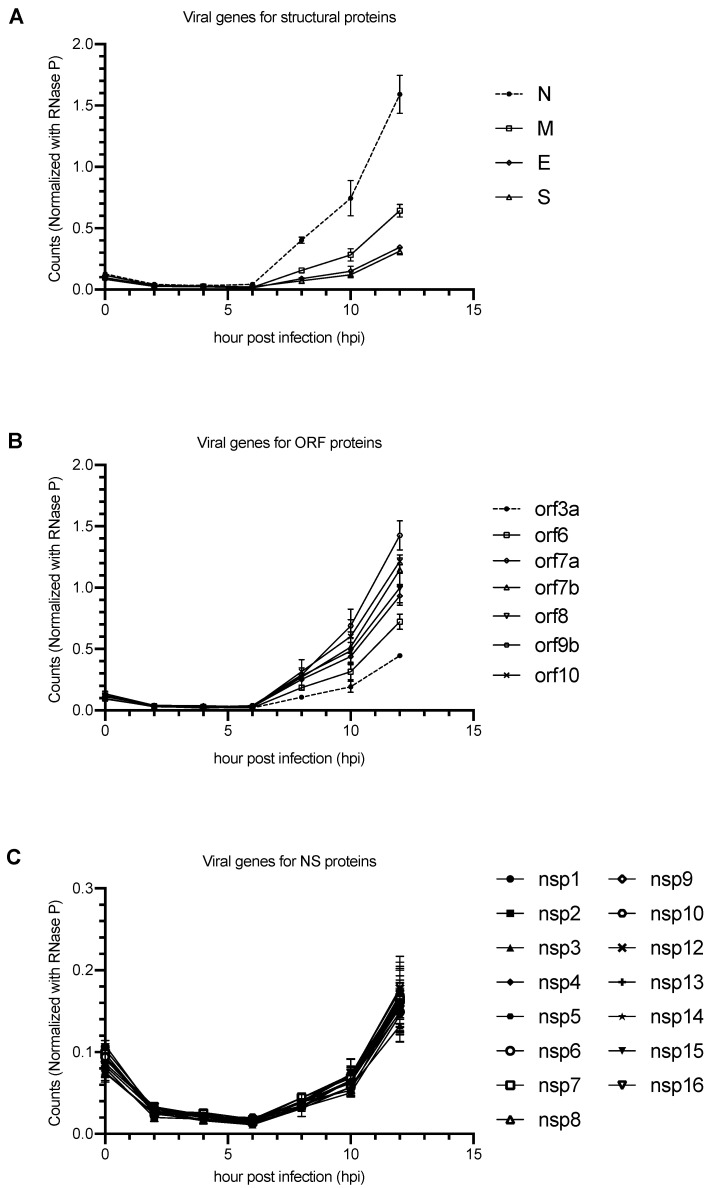
Transcription profiles of different viral genes of SARS-CoV-2 in Caco-2 cells. Human Caco-2 cells were infected by the SARS-CoV-2 at a moi of 0.01 and the total RNA was collected at 15 min (0), and 2, 4, 6, 8, 10, and 12 h after infection. The transcription levels of the viral genes were determined by ddPCR using corresponding primers and probe. (**A**) Structural genes, (**B**) ORF genes, (**C**) NSP genes. All counts were normalized with endogenous control (RNase P gene).

**Figure 3 pathogens-10-01274-f003:**
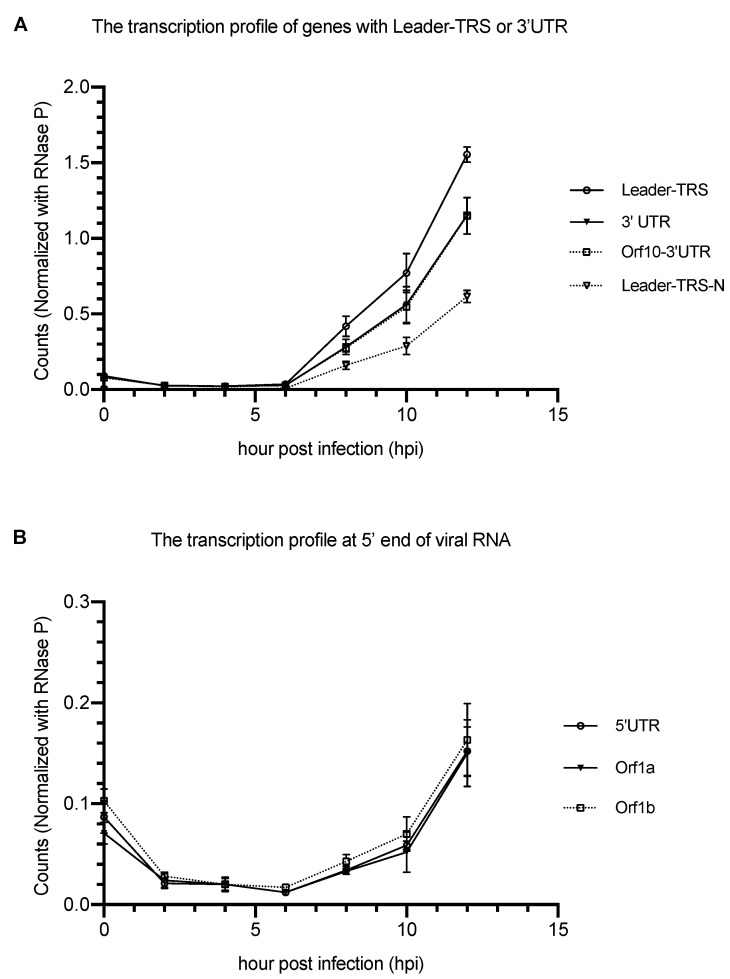
Transcription profiles at the 3′ and 5′ end of the viral RNA in Caco-2 cells. Human Caco-2 cells were infected by the SARS-CoV-2 at a moi of 0.01 and the total RNA was collected at 15 min (0), and 2, 4, 6, 8, 10, and 12 h after infection. The transcription levels of the viral genes were determined by ddPCR using corresponding primers and probe. (**A**) The transcription quantification of the Leader–TRS, 3′ UTR, Orf10-3′ UTR and Leader-TRS-N. (**B**) The transcription quantification of orf1a (nsp2–3), orf1b (nsp12–13), and the 5′ UTR. All the counts were normalized with endogenous control (RNase P gene).

**Table 1 pathogens-10-01274-t001:** Primers and probes for ddPCR.

Gene Categories	Target Regions	Primer/Probe	Sequence (5′ to 3′)	Position	Amplicon Size
**Structural Proteins**	S (A)	Forward	GTGACATCTCTGGCATTAATGC	25062–25173	112
		Reverse	CCAAGTTCTTGGAGATCGATGAG		
		Probe	TGGCAACCTCATTGAGGCGGTC		
	E	Forward	GGTACGTTAATAGTTAATAGCGTAC	26272–26395	124
		Reverse	GACTCACGTTAACAATATTGCAG		
		Probe	TCCTTACTGCGCTTCGATTGTGTG		
	M	Forward	GTGGACATCTTCGTATTGCTG	26959–27081	123
		Reverse	CACGCTGCGAAGCTCCCAA		
		Probe	CAACAGTGATTTCTTTAGGCAGGTCC		
	N (A)	Forward	GAAGTCACACCTTCGGGAAC	29240–29323	84
		Reverse	GACTTGATCTTTGAAATTTGGATCT		
		Probe	TGGTTGACCTACACAGGTGCCATC		
**Nonstructural Proteins (NSP)**	orf1a (nsp2–3)	Forward	CCCTTGCACCTAATATGATGG	2667–2762	96
		Reverse	CTTCTATCACAGTGTCATCACC		
		Probe	CTCAAAGGCGGTGCACCAACAAAG		
	orf1b (nsp12–13)	Forward	CACTTCAAGGTATTGGGAACC	16173–16304	132
		Reverse	GGTCTACGTATGCAAGCACC		
		Probe	CAGTCTTACAGGCTGTTGGGGCTT		
	nsp1	Forward	TTCAACGAGAAAACACACGTCC	287–407	121
		Reverse	CTTTAAGATGTTGACGTGCCTC		
		Probe	CTTTGGAGACTCCGTGGAGGAGG		
	nsp2	Forward	GTATTAACGGGCTTATGTTGCTC	2616–2706	91
		Reverse	GTGAAGGTATTGTTTGTTACCATC		
		Probe	CAGAAAAGTACTGTGCCCTTGCACC		
	nsp3 (A)	Forward	GACATAGAAGTTACTGGCGATAG	8249–8367	119
		Reverse	GCATTAATATGACGCGCACTAC		
		Probe	CATGACACCCCGTGACCTTGG		
	nsp4	Forward	GCTACAGAGAAGCTGCTTGTT	9939–10051	113
		Reverse	CAAAACAGCTGAGGTGATAGAG		
		Probe	CATCAGAACCTGAGTTACTGAAGTC		
	nsp5	Forward	GGAGTTCATGCTGGCACAGA	10562–10685	124
		Reverse	CAGCGTACAACCAAGCTAAAAC		
		Probe	ACAAGCAGCTGGTACGGACACAAC		
	nsp6	Forward	GTGTTATGTATGCATCAGCTGTAG	11310–11404	95
		Reverse	ATTCATAAGTGTCCACACTCTCC		
		Probe	CACCATCATCATACACAGTTCTTGC		
	nsp7	Forward	GTCAGATGTAAAGTGCACATCAG	11851–11945	95
		Reverse	ACTGGACACATTGAGCCCACA		
		Probe	CTCAGTTTTGCAACAACTCAGAGTAG		
	nsp8	Forward	GGCTAAATCTGAATTTGACCGTG	12223–12296	74
		Reverse	GGGTCATAGCTTGATCAGCC		
		Probe	CCAACTTACGTTGCATGGCTGCA		
	nsp9	Forward	CTAAGAGTGATGGAACTGGTAC	12855–12938	84
		Reverse	CTTTAGGACCTTTAGGTGTGTCT		
		Probe	CCTACAAGGTGGTTCCAGTTCTG		
	nsp10	Forward	TGCTGTAGATGCTGCTAAAGCT	13025–13169	88
		Reverse	TGTGTGTACACAACATCTTAACAC		
		Probe	TGGTTGTCCCCCACTAGCTAGA		
	nsp12	Forward	GTCATGTGTGGCGGTTCACT	15439–15510	72
		Reverse	AGCATAAGCAGTTGTGGCATC		
		Probe	CCTGATGAGGTTCCACCTGGTTTAAC		
	nsp13	Forward	CTATAGGTCCAGACATGTTCCTC	17528–17602	75
		Reverse	CCAAAGCACTCACAGTGTCAAC		
		Probe	CAGCAGGACAACGCCGACAAGTTC		
	nsp14	Forward	GTATAACACGTTGCAATTTAGGTG	19457–19604	148
		Reverse	GTGTTCCAGAGGTTATAAGTATC		
		Probe	TCAGCTGGCTTTAGCTTGTGGGTT		
	nsp15	Forward	GCATTTGAGCTTTGGGCTAAGC	19780–19861	82
		Reverse	CAGCAATGTCCACACCCAAAT		
		Probe	CAACATTAAACCAGTACCAGAGGTG		
	nsp16	Forward	CAGGTACAGCTGTTTTAAGACAG	20897–20977	81
		Reverse	CATCAGAGACAAAGTCATTAAGATC		
		Probe	CAGCGTACCCGTAGGCAACC		
**Accessory Proteins**	orf3a	Forward	CAAGGTGAAATCAAGGATGCTAC	25441–25517	77
		Reverse	GGGAGTGAGGCTTGTATCGG		
		Probe	CTTCAGATTTTGTTCGCGCTACTGC		
	orf6	Forward	GTTTCATCTCGTTGACTTTCAGG	27204–27289	86
		Reverse	CAAGATTCCAAATGGAAACTTTAAAAG		
		Probe	CCTCATAATAATTAGTAATATCTCTGC		
	orf7a	Forward	GCTTTAGCACTCAATTTGCTTTTGC	27566–27640	75
		Reverse	AAACTGATCTGGCACGTAACTG		
		Probe	TGTCCTGACGGCGTAAAACACGTC		
	orf7b	Forward	GCTTTTTAGCCTTTCTGCTATTCC	27790–27884	95
		Reverse	GGCGTGACAAGTTTCATTATGATC		
		Probe	CTTTTGGTTCTCACTTGAACTGC		
	orf8	Forward	CAGCACCTTTAATTGAATTGTGC	28054–28193	140
		Reverse	CACTACAAGACTACCCAATTTAGG		
		Probe	CCCATTCAGTACATCGATATCGG		
	orf9b	Forward	CCCAATAATACTGCGTCTTGG	28409–28492	84
		Reverse	TGGAACGCCTTGTCCTCGAG		
		Probe	CACCGCTCTCACTCAACATGGC		
	orf10	Forward	TGGGCTATATAAACGTTTTCGCT	29559–29642	84
		Reverse	GTGCTATGTAGTTACGAGAATTC		
		Probe	CCGTTTACGATATATAGTCTACTC		
**Others**	Leader–TRS	Forward	TTAAAGGTTTATACCTTCCCAGG	2–75	74
		Reverse	GTTCGTTTAGAGAACAGATCTAC		
		Probe	AACAAACCAACCAACTTTCGATCTCT		
	5′ UTR	Forward	GACAGGACACGAGTAACTCG	155–229	75
		Reverse	TGCTGATGATCGGCTGCAAC		
		Probe	CTGCAGGCTGCTTACGGTTTCG		
	3′ UTR	Forward	CACCACATTTTCACCGAGGC	29719–29795	77
		Reverse	CCATATAGGCAGCTCTCCC		
		Probe	CTGTACACTCGATCGTACTCCGC		
	Leader–TRS–N	Forward	CCCAGGTAACAAACCAACCAAC	19–28332	N/A
		Reverse	GGTCCACCAAACGTAATGCG		
		Probe	CCCCAAAATCAGCGAAATGCACC		
	orf10-3′ UTR	Forward	GAATTCTCGTAACTACATAGCAC	29620–29743	124
		Reverse	GCGTGGCCTCGGTGAAAATG		
		Probe	CATTAGGGAGGACTTGAAAGAGCC		
	S (B)	Forward	GTTCTTGTGGATCCTGCTGC	25305–25378	74
		Reverse	GTAATGTAATTTGACTCCTTTGAGC		
		Probe	TGATGAAGACGACTCTGAGCCAG		
	N (B)	Forward	CTCATCACGTAGTCGCAACAG	28831–28940	110
		Reverse	GCAGCAAAGCAAGAGCAGCA		
		Probe	CCTGCTAGAATGGCTGGCAATGGC		
	nsp3 (B)	Forward	CGTTAAAGATTTCATGTCATTGTCTG	8407–8513	107
		Reverse	CTTGTCTAGTAGTTGCACATGTC		
		Probe	CTACGAAAACAAATACGTAGTGCTGCT		
**Endogenous Controls**	RNase P	Forward	AGATTTGGACCTGCGAGCG	28–114	87
		Reverse	GCAACAACTGAATAGCCAAGG		
		Probe	TTCTGACCTGAAGGCTCTGCGCG		

S(B), N(B), and nsp3(B): the alternative sets of primers/probes.

**Table 2 pathogens-10-01274-t002:** CT value of the clinical specimens using the primers and probe designed in this study.

	Sample (CT Value)											
	1	2	3	4	5	6	7	8	9	10	11	12
**S**	14.6	16.9	15.9	25.1	24.5	25.8	26.6	26.4	19.5	16.6	15.7	13.7
**E**	17.5	19.9	18.4	28.1	27.7	28.8	29.9	29.4	22.3	19.3	18.0	16.5
**M**	17.7	19.6	19.1	27.8	26.1	29.0	29.8	29.5	22.3	19.5	18.8	16.9
**N**	12.8	15.4	15.5	22.8	23.6	24.6	25.1	25.4	17.1	13.9	13.3	11.9
**orf1a**	14.4	16.5	15.6	24.3	24.5	24.9	26.5	26.3	19.7	16.7	15.8	13.4
**orf1b**	13.4	16.2	15.7	24.0	23.7	25.1	25.6	25.6	19.3	16.4	15.5	13.4
**nsp1**	14.0	16.6	15.6	24.4	24.2	25.6	26.0	26.4	19.8	16.9	16.0	13.7
**nsp2**	14.9	17.4	16.6	25.2	24.4	26.0	26.3	26.8	20.3	17.5	16.8	14.2
**nsp3**	14.8	17.2	16.7	24.9	23.7	26.0	26.0	26.6	19.9	17.3	16.4	13.9
**nsp4**	15.8	18.0	18.0	26.3	24.4	26.9	27.6	27.5	21.1	18.3	17.5	15.1
**nsp5**	15.4	18.0	18.1	25.6	24.4	26.8	27.6	27.8	21.1	18.4	17.6	15.2
**nsp6**	18.5	19.5	19.6	28.0	28.3	28.7	30.5	30.5	23.7	20.5	20.9	17.7
**nsp7**	15.5	17.6	16.8	25.4	24.5	25.9	27.2	26.9	20.9	17.9	17.3	15.0
**nsp8**	14.3	17.1	16.3	24.9	24.4	25.5	26.5	26.3	20.2	17.4	16.8	14.2
**nsp9**	15.0	17.4	16.6	25.4	25.2	26.3	26.8	27.3	20.6	17.9	17.3	14.7
**nsp10**	14.9	17.6	17.3	25.4	25.4	26.5	27.2	27.5	20.8	18.0	17.2	14.7
**nsp12**	14.9	17.7	17.4	25.6	25.0	26.2	27.1	26.5	20.5	18.1	17.5	14.8
**nsp13**	14.0	16.6	16.3	24.4	24.0	24.9	26.2	27.2	19.7	17.0	16.2	13.9
**nsp14**	14.1	16.8	16.6	24.5	24.2	25.4	26.2	26.2	19.8	17.0	16.2	13.9
**nsp15**	15.6	18.0	17.5	25.9	25.8	26.6	27.4	27.2	21.1	18.4	17.5	15.3
**nsp16**	13.6	16.4	15.9	24.2	23.5	24.6	25.8	25.4	19.4	16.4	15.6	13.6
**orf3a**	15.3	17.3	15.6	25.1	25.6	25.8	27.4	27.4	19.5	16.5	15.8	13.9
**orf6**	16.8	18.9	18.0	27.2	28.1	28.9	29.9	29.6	21.0	18.1	17.7	15.6
**orf7a**	13.3	16.2	15.4	24.0	24.5	25.2	26.2	25.8	18.3	15.0	14.4	12.7
**orf7b**	13.8	15.9	14.8	23.9	24.3	25.1	25.9	25.6	17.9	14.8	14.6	12.3
**orf8**	14.2	16.5	15.2	24.1	24.6	25.3	26.5	25.9	20.2	17.2	16.5	14.9
**orf9b**	12.8	14.7	13.9	22.6	23.5	23.7	25.3	24.8	15.7	12.9	13.0	10.6
**orf10**	16.0	17.9	16.9	25.7	27.3	27.2	29.2	28.5	19.8	16.4	16.0	14.6

## Data Availability

All data are reported in the manuscript and the Appendix A.

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
