# Peer review of "Tracking the Transcription Kinetic of SARS-CoV-2 in Human Cells by Reverse Transcription-Droplet Digital PCR"

_pathogens, 2021, doi:10.3390/pathogens10101274_

Round 1

Reviewer 1 Report

The manuscript by Au et al. investigated the kinetics of transcription of viral genes in Caco-2 cells infected by SARS-CoV-2.   The results would be of value for helping us to understand the life cycle of SARS-CoV-2.

I think this manuscript will be stronger if the following points can be explored:

  1. The authors need to check the virus titers in the supernatant in the medium after infecting Caco-2.
  2. The authors had better to check the expression level of respective protein corresponding to the mentioned genes by Western blotting to confirm the results of ddPCR.

Author Response

Reviewer 1

The manuscript by Au et al. investigated the kinetics of transcription of viral genes in Caco-2 cells infected by SARS-CoV-2. The results would be of value for helping us to understand the life cycle of SARS-CoV-2.

I think this manuscript will be stronger if the following points can be explored:

  1. The authors need to check the virus titers in the supernatant in the medium after infecting Caco-2.

Response: We thank for the suggestion from the reviewer. Caco-2 cells have been shown to susceptible for the replication of SARS-CoV-2. In this study, our aim was to monitor the transcription kinetic of SARS-CoV-2 within one virus life cycle. To avoid the noise from the background, low virus input (MOI=0.01) was used. Thus, a longer incubation period will be needed in order to detect a significant virion production in the supernatant after multiple replication cycles (>12 hours post-infection) under our experimental setting.

  1. The authors had better to check the expression level of respective protein corresponding to the mentioned genes by Western blotting to confirm the results of ddPCR.

Response: While SARS-COV-2 is a newly identified pathogen, the reagents which are used for the detection of different viral proteins especially the NSPs and ORFs are not available in the market. In addition, the expressions of the NSPs and ORFs are expected to be low and hard to be visualized by Western Blotting under our experimental condition.

Reviewer 2 Report

The authors in this manuscript have designed specific primers/probe sets to detect the RNA level of SARS-COV-2 encoded 15 NSP, 7 ORF and 4 structural genes using ddPCR method. The authors were able to detect all the transcripts in human Caco-2 cell line. 

Showing that these primers/probes can be used to detect virus in the patient samples would have strengthen the manuscript.

Author Response

Reviewer 2

The authors in this manuscript have designed specific primers/probe sets to detect the RNA level of SARS-COV-2 encoded 15 NSP, 7 ORF and 4 structural genes using ddPCR method. The authors were able to detect all the transcripts in human Caco-2 cell line.

Showing that these primers/probes can be used to detect virus in the patient samples would have strengthen the manuscript.

Response: We have tested 12 clinical specimens using our primer/probe sets which target to the structural protein, NSP or ORF. We showed that all the primers/probes can efficiently amplify the corresponding target genes. The results have been added into the manuscript.

Reviewer 3 Report

The manuscript entitled “Tracking the Transcription Kinetic of SARS-CoV-2 in human cell by Reverse Transcription-Droplet Digital PCR” by Ka Ki Au and colleagues seeks to advance the understanding of SARS-CoV-2 transcriptional dynamics via ddPCR in order to assist with the development of novel viral replication therapies. To this end, they have designed sets of primers and probes to measure the expression levels of 15 non-structural proteins, 7 open reading frames and 4 structural genes (spike, envelope, matrix and nucleocapsid), which they employ in a Caco-2 cell culture infection system (immortalized human colorectal adenocarcinoma). The authors estimate that it takes around 6 hours to initiate viral transcription following infection.  

This study is very well designed and executed. For example, the authors take the time to perform alignments of many existing SARS-CoV-2 sequences to identify the most variable areas and exclude them from their primer designs. This means that even though the primers were based on sequences downloaded around a year ago, they stand the best chance at remaining relevant. The methods are very detailed and address every important aspect of their work. The primers are shared as part of Table 1. Similarly, the discussion addresses the questions that arise as one reads the manuscript, for example, specific variants of concern and how they would perform with this assay, comparison to similar studies (other cell lines, methods), interpretation of the uncovered dynamics, and comparison to other related viruses.  

Major 

  • No major concerns 

Minor 

  • Some areas of the manuscript need English editing. However, it is very clearly written and easily understood so not a major concern. 
  • It would be interesting to get the authors’ perspective on how remdesivir would affect the dynamics they report (ideally it would be interesting to see those experiments but that would be a separate manuscript). Since the purported goal is to create a system and set of baseline results that would assist in developing new treatments targeting SARS-CoV-2 replication, it would be helpful to briefly expand on how this culture system could be used to test said therapies, and which potential therapies may be in development at this time that would be good candidates. This could all be addressed in one additional, small paragraph in the discussion. 

Overall, this is a well-designed and well-executed study that advances the understanding of early viral dynamics upon cell infection in a well-controlled culture system. The authors clearly describe what they do and their reasoning and do not overstate their claims. In order to bring this study to the next level, one would include remdesivir as well as other potential novel therapies. However, this additional work can be explored and reported as a separate manuscript.

Author Response

Reviewer 3

The manuscript entitled “Tracking the Transcription Kinetic of SARS-CoV-2 in human cell by Reverse Transcription-Droplet Digital PCR” by Ka Ki Au and colleagues seeks to advance the understanding of SARS-CoV-2 transcriptional dynamics via ddPCR in order to assist with the development of novel viral replication therapies. To this end, they have designed sets of primers and probes to measure the expression levels of 15 non-structural proteins, 7 open reading frames and 4 structural genes (spike, envelope, matrix and nucleocapsid), which they employ in a Caco-2 cell culture infection system (immortalized human colorectal adenocarcinoma). The authors estimate that it takes around 6 hours to initiate viral transcription following infection. 

This study is very well designed and executed. For example, the authors take the time to perform alignments of many existing SARS-CoV-2 sequences to identify the most variable areas and exclude them from their primer designs. This means that even though the primers were based on sequences downloaded around a year ago, they stand the best chance at remaining relevant. The methods are very detailed and address every important aspect of their work. The primers are shared as part of Table 1. Similarly, the discussion addresses the questions that arise as one reads the manuscript, for example, specific variants of concern and how they would perform with this assay, comparison to similar studies (other cell lines, methods), interpretation of the uncovered dynamics, and comparison to other related viruses. 

Major

  • No major concerns

Response: We appreciate to the positive comments from the reviewer.

Minor

  • Some areas of the manuscript need English editing. However, it is very clearly written and easily understood so not a major concern.

Response: We have checked the grammatical mistakes in the manuscript as suggested.

  • It would be interesting to get the authors’ perspective on how remdesivir would affect the dynamics they report (ideally it would be interesting to see those experiments but that would be a separate manuscript). Since the purported goal is to create a system and set of baseline results that would assist in developing new treatments targeting SARS-CoV-2 replication, it would be helpful to briefly expand on how this culture system could be used to test said therapies, and which potential therapies may be in development at this time that would be good candidates. This could all be addressed in one additional, small paragraph in the discussion. Overall, this is a well-designed and well-executed study that advances the understanding of early viral dynamics upon cell infection in a well-controlled culture system. The authors clearly describe what they do and their reasoning and do not overstate their claims. In order to bring this study to the next level, one would include remdesivir as well as other potential novel therapies. However, this additional work can be explored and reported as a separate manuscript.

Response: We have added a paragraph to describe the potential benefit of using our model to study antiviral drugs against SARS-CoV-2 in the coming future. “While the efficacy of remdesivir is still sub-optimal, structurally modification of this drug or identification of new compound will be one of the key directions for antiviral research. As we have demonstrated the kinetic of the transcription process of the SARS-CoV-2 in caco-2 cells, our model can help to investigate the functions of the antiviral drugs such as extending the incubation time before the initiation of viral transcription or reducing the transcription level etc. Our study thus provides a model for sensitively evaluating the performance of any new antiviral drugs against the SARS-CoV-2 and their mechanism of action.”